# Oscillations Make Neural Networks Robust to Quantization

**Jonathan Wenshøj**                                                          *jowe@di.ku.dk*
**Bob Pepin**                                                                 *bope@di.ku.dk*
**Raghavendra Selvan**                                                        *raghav@di.ku.dk*
*Department of Computer Science, University of Copenhagen, Denmark*

**Reviewed on OpenReview:** *https://openreview.net/forum?id=bPwcJOnkDC*

## Abstract

We challenge the prevailing view that weight oscillations observed during Quantization Aware Training (QAT) are merely undesirable side-effects and argue instead that they are an essential part of QAT. We show in a univariate linear model that QAT results in an additional loss term that causes oscillations by pushing weights *away* from their nearest quantization level. Based on the mechanism from the analysis, we then derive a regularizer that induces oscillations in the weights of neural networks during training. Our empirical results on ResNet-18 and Tiny Vision Transformer, evaluated on CIFAR-10 and Tiny ImageNet datasets, demonstrate across a range of quantization levels that training with oscillations followed by post-training quantization (PTQ) is sufficient to recover the performance of QAT in most cases. With this work we provide further insight into the dynamics of QAT and contribute a novel insight into explaining the role of oscillations in QAT which until now have been considered to have a primarily negative effect on quantization.[1]

## 1 Introduction

Deep neural networks have grown increasingly successful at solving difficult modelling tasks, but are also increasingly becoming more expensive to use. As model sizes balloon into the hundreds of millions of parameters, the cost of inference has become a significant bottleneck, particularly for deployment on edge devices or usage in large-scale services (Sevilla et al., 2022). To reduce this cost, quantization of model weights has emerged as a prominent strategy (Nagel et al., 2021).

Weight quantization, however, comes with its own cost, namely a drop in accuracy. Reducing precision introduces quantization error, and naive approaches like Post-Training Quantization (PTQ) often lead to sharp degradations in model performance at low bit widths. To combat this degradation in accuracy, many methods have been proposed, usually centered around minimizing the quantization error (Hung et al., 2015; Hirose et al., 2017; Li et al., 2020; Choi et al., 2020; Han et al., 2021; Zhong et al., 2025).

Yet these approaches fall short of Quantization-Aware Training (QAT) (Jacob et al., 2018; Krishnamoorthi, 2018), which integrates quantization directly into the training loop, yielding models that remain robust even under low-precision constraints. However, despite its empirical success, the underlying behavior of QAT remains poorly understood. In particular, QAT often exhibits *oscillations* in the weights during optimization, while regular training does not. Rather than settling into a stable state, quantized weights fluctuate between adjacent quantization levels. These oscillations are widely seen as *undesirable artifacts.* Several works have tried to suppress these weight oscillations mainly through dampening or weight freezing (Défossez et al., 2022; Nagel et al., 2022; Huang et al., 2024; Gupta & Asthana, 2024; Liu et al., 2023).

This paper poses the question: *Have we misunderstood the nature of oscillations in QAT?* To the best of our knowledge, we are the first to argue that weight oscillations in QAT are not a flaw, but rather an essential feature of the training dynamics. Using a toy model, we show that oscillations arise from a

---

[1]Source code is available at https://github.com/saintslab/osc_reg.

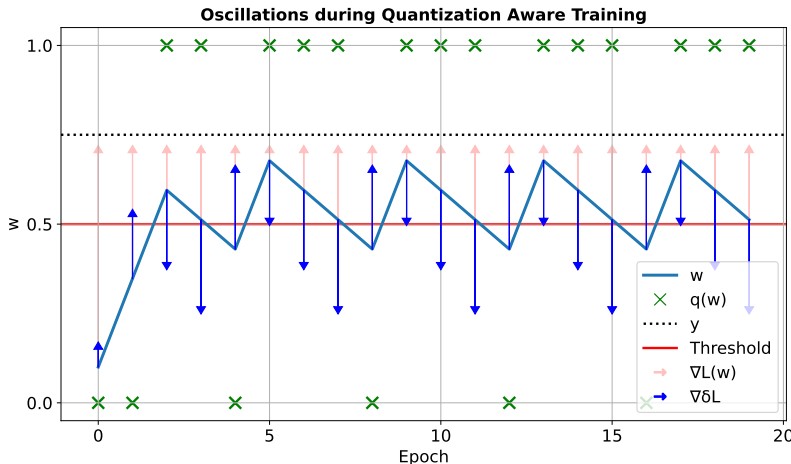

Figure 1: Oscillatory behavior during QAT in a one weight linear model $\hat{y} = q(w)x$, with weight $w$, quantized weight $q(w)$, input $x = 1$ and target $y = 0.75$. The gradient of the loss term of this model during QAT can be decomposed into two terms: $\nabla L(w)$ and $\nabla \delta_L = q(w) - w$, where the latter term is what differentiates QAT from just optimizing the full-precision loss $L(w)$. During QAT, $\nabla L(w)$ always points towards $y$, while $\nabla \delta_L$ introduces a dynamic which pushes $w$ towards the nearest bin threshold. This causes $w$ to oscillate when $y$ is not exactly on a quantization level. In the above case this makes $q(w)$ alternate between 0 and 1. Note the frequency of oscillations of $q(w)$ lets the quantized weight on average to converge to 0.75.

gradient component in QAT which pushes weights towards the nearest quantization threshold – this stands in contrast to aligning weights at the quantization level, which would minimize the quantization error. We then induce such oscillations to push the weights towards the nearest thresholds deliberately during the model training. This is realized as a simple regularization term that pushes weights towards their nearest quantization threshold. Surprisingly, this leads to models that are robust to quantization and in most cases we find that these regularization-induced oscillations recover the accuracy obtained with QAT. Additionally, in many cases, oscillations outperform QAT under cross-bit evaluation (i.e. testing at precision levels not simulated during training). These findings further deepen our understanding of QAT and point to a more nuanced role for weight oscillations; suggesting they may be beneficial to QAT rather than being solely detrimental as argued in most existing literature.

Our primary contributions supporting this claim are:

1. Using a univariate model we illustrate how the Straight Through Estimator (STE) leads to oscillations and clustering of model weights during QAT (Sec. 3);

2. We show experimentally that by using a mechanism inspired by the toy model, we can induce oscillations and clustering during training in neural networks (Sec. 4).

3. We empirically confirm using the CIFAR-10 (Krizhevsky & Hinton, 2009) and Tiny-ImageNet (Le & Yang, 2015) datasets on a multi-layer perceptron (MLP), ResNet-18 (He et al., 2016) and Tiny Vision transformer (Tiny ViT) (Wu et al., 2022) that introducing oscillations through regularization in most cases recovers the accuracy of QAT (Sec. 5).

## 2  Preliminaries and Related Work

**Quantization:**  A quantizer divides a continuous input range into quantization bins, where each bin is represented by a specific quantization level. The boundaries between bins are called quantization thresholds. During quantization, any value within a bin is mapped to that bin's quantization level. With a uniform quantizer, the step size (the distance between two adjacent quantization levels) is equal to the scale factor $s$.

We consider a uniform symmetric, scalar quantizer, $q(\cdot)$, that can then be expressed as:

$$q(\mathbf{w}) = s \cdot \left\lceil \frac{\mathbf{w}}{s} \right\rfloor. \tag{1}$$

Here, $s$ represents the scale factor and $\lceil \cdot \rfloor$ denotes the rounding operation.

The scale factor $s$ is set to cover the range of $\mathbf{w}$ as this removes the need for the clamping operation $\text{clamp}(q(\mathbf{w}); \alpha, \beta)$ in the quantizer. The function clamp(.) restricts $\alpha \leq q(w_i) \leq \beta$ and any values less or greater than is set to $\alpha$ and $\beta$ respectively. Additionally we set the number of positive and negative quantization levels to be $2^{b-1} - 1$, so we have symmetric number of levels around 0:

$$s = \frac{\max(|\mathbf{w}|)}{2^{b-1} - 1}, \tag{2}$$

where $b$ is the number of bits the quantizer can use.

The quantization process introduces quantization error $\boldsymbol{\Delta}$, defined as the difference between the original and quantized values:

$$\boldsymbol{\Delta}(\mathbf{w}) = \mathbf{w} - q(\mathbf{w}). \tag{3}$$

Due to the uniform quantizer each component of the absolute error (due to quantizing $w_i$) is bounded between $0 \leq |\boldsymbol{\Delta}_i| \leq s/2$, for all the bins. This is maximized at quantization thresholds and is 0 at quantization levels.

**Straight-Through Estimator (STE):** STE is a heuristic gradient approximation technique commonly used to enable backpropagation through functions $f(\cdot)$ whose gradients are zero almost everywhere (Bengio et al., 2013). In the forward pass, the function $f(\mathbf{x})$ is applied as usual. During the backward pass, however, its gradient is approximated by replacing the derivative of $f$ with the constant 1. Formally, STE defines the approximate gradient of a function $f$ as

$$\frac{\partial f}{\partial \mathbf{x}} \approx \frac{\hat{\partial} f}{\hat{\partial} \mathbf{x}} = 1. \tag{4}$$

**Quantization-Aware Training (QAT):** In QAT, the network maintains full-precision weights for learning but quantizes them during the forward pass before computation. This allows training to proceed with gradients applied to full-precision weights, while the network effectively operates on quantized values. While there exist many variants of QAT, the typical forward pass is performed using the quantized weights $q(\mathbf{w})$ (Jacob et al., 2018; Krishnamoorthi, 2018). The gradient for the weights during QAT is given by

$$\frac{\partial \mathcal{L}(q(\mathbf{w}))}{\partial \mathbf{w}} = \frac{\partial \mathcal{L}(q(\mathbf{w}))}{\partial q(\mathbf{w})} \cdot \frac{\partial q(\mathbf{w})}{\partial \mathbf{w}}. \tag{5}$$

A problem with the above formulation is that the gradient of the quantizer $\frac{\partial q(\mathbf{w})}{\partial \mathbf{w}}$ is zero almost everywhere, causing the last term to interrupt gradient-based learning. A widely used solution to this problem is to use the STE (Eq. (4)).

**Weight Oscillations:** We use the definition from Nagel et al. (2022) which defines a weight oscillation during QAT to occur in iteration $t$ if it satisfies both the following conditions:

1. The quantized value of the weight needs to change between iterations i.e. $q(w_t) \neq q(w_{t-1})$.

2. The direction of the change in the quantized domain needs to be the opposite than that of its previous change i.e. $\text{sign}(\Gamma_t) \neq \text{sign}(\Gamma_\tau)$, where $\tau$ is the iteration of the last change, and $\Gamma_t = q(w_t) - q(w_{t-1})$ is the direction of the change.

**Related Work:** Minimizing the quantization error is the most commonly used strategy to reduce the impact of quantization on model accuracy. Extensive research has been dedicated to developing techniques that explicitly minimize the quantization error during optimization (Hung et al., 2015; Hirose et al., 2017; Li et al., 2020; Choi et al., 2020; Han et al., 2021; Zhong et al., 2025). Despite these efforts, the aforementioned strategies often fall short of the accuracy obtained with QAT (Jacob et al., 2018) at individual bits or indirectly rely upon QAT themselves.

However, there is limited understanding of how QAT affects model optimization and why it outperforms other methods. One phenomenon observed during QAT is weight oscillations (Défossez et al., 2022; Nagel et al., 2022), which are periodic changes in the value of the quantized weight between two adjacent quantization levels. It is speculated in these works that the abrupt changes in values caused by oscillations can interfere negatively with optimization. Oscillations are assumed to be undesirable side effects caused by the use of the STE during backpropagation, as the STE allows gradients to pass through the rounding operation in the quantizer, which has a gradient of zero almost everywhere (Défossez et al., 2022; Nagel et al., 2022).

Several approaches have been suggested to mitigate oscillations by either freezing or dampening (Défossez et al., 2022; Nagel et al., 2022; Huang et al., 2024; Gupta & Asthana, 2024; Liu et al., 2023). However, the reported accuracy gains are sometimes marginal, and these methods may inadvertently also hinder the optimization process. For instance, Nagel et al. (2022) notes that freezing or dampening weights too early during training can hurt optimization, indicating that oscillations might contribute to finding better quantized minima of the loss. Liu et al. (2023) propose that weights with low oscillation frequency should be frozen, where as high-frequency ones should be left unfrozen, under the rationale that high frequency means the network has little confidence in what value to quantize the weight to, whereas low frequency means the optimal weight lies close to a quantization level.

## 3   Oscillations in QAT

Previous studies have explored linear models to analyze the behavior of QAT and the phenomenon of weight oscillations (Défossez et al., 2022; Nagel et al., 2022; Liu et al., 2023; Gupta & Asthana, 2024). Inspired by these works, we also analyze a linear regression model to gain insights into the optimization dynamics during QAT.

### 3.1   Toy Model

Consider a linear model with a single weight $w$, input $x$ and target $y \in \mathbb{R}$. The quantized version of this model is defined as $\hat{y} = q(w)x$, where $q(\cdot)$ is the quantizer from Eq. (1). The quadratic loss for the quantized model is given by

$$\mathcal{L}(q(w)) = \frac{1}{2}(\hat{y} - y)^2 = \frac{1}{2}(q(w)x - y)^2. \tag{6}$$

Our goal in this section is to understand how QAT affects the full-precision optimization process. For a given loss function $\mathcal{L}(\cdot)$ with quantized weights, we have

$$\mathcal{L}(q(w)) = \mathcal{L}(w) + \mathcal{L}(q(w)) - \mathcal{L}(w). \tag{7}$$

We can then expand the difference in loss caused by quantization as follows

$$\delta_{\mathcal{L}} = \mathcal{L}(q(w)) - \mathcal{L}(w) = \frac{1}{2}\left((q(w)x - y)^2 - (wx - y)^2\right) \tag{8}$$

$$= \frac{1}{2}\left(x^2\left(q(w)^2 - w^2\right)\right) + (yx(w - q(w))). \tag{9}$$

This expression decomposes the loss difference $\delta_{\mathcal{L}}$ into a quadratic term $\frac{1}{2}x^2(q(w)^2 - w^2)$ and a linear term $yx(w - q(w))$.

Next we derive the gradient of $\delta_{\mathcal{L}}$ wrt. $w$

$$\frac{\partial \delta_{\mathcal{L}}}{\partial w} = \frac{\partial}{\partial w}\left(\mathcal{L}(q(w)) - \mathcal{L}(w)\right) = \frac{\partial}{\partial w}\left(\frac{1}{2}x^2(q(w)^2 - w^2) + yx(w - q(w))\right) \tag{10}$$

$$= x^2\left(q(w)\frac{\partial q(w)}{\partial w} - w\right) + yx\left(1 - \frac{\partial q(w)}{\partial w}\right). \tag{11}$$

Using the STE and recalling that $\frac{\hat{\partial} q}{\hat{\partial} w} = 1$ the expression of the STE gradient simplifies to[2]

$$\frac{\hat{\partial} \delta_{\mathcal{L}}}{\hat{\partial} w} = x^2(q(w) - w) = -x^2 \mathbf{\Delta}(w). \tag{12}$$

### 3.2 Oscillation Mechanism

To see how the observations in Sec. 3.1 give rise to oscillations, let $w_0$ denote the upper quantization threshold for an arbitrary weight $w$, defined as $w_0 = q(w) + s/2$. For $\varepsilon \in (0, s/2)$, note that we have $q(w_0 - \varepsilon) = q(w)$ and $q(w_0 + \varepsilon) = q(w) + s$, so that

$$\mathbf{\Delta}(w_0 + \varepsilon) = q(w) + s/2 + \varepsilon - (q(w) + s) = -s/2 + \varepsilon, \tag{13}$$
$$\mathbf{\Delta}(w_0 - \varepsilon) = q(w) + s/2 - \varepsilon - q(w) = s/2 - \varepsilon. \tag{14}$$

Assuming $x \neq 0$, the negative STE gradient "flips" from $-s/2$ to $+s/2$ as the weight $w$ passes the quantization threshold $w_0$, pushing the weight back towards the threshold. We note that the STE gradient is zero at the special value $w = q(w)$, but the preceding argument shows that this is an unstable critical point and gradient noise will immediately cause the weights to move away from it. When combined with (stochastic) gradient descent and a finite discretization timestep we can identify this as the driving mechanism behind oscillations during training with QAT (Fig. 1).

### 3.3 Weight Clustering

We can also see how the dynamics described above can lead to weight clustering around quantization thresholds by looking at the sign of $\mathbf{\Delta}$ for different values of $w$. For a weight $w$ let $d_{\text{low}}(w)$ and $d_{\text{up}}(w)$ denote the distance from $w$ to the upper and lower thresholds, $d_{\text{low}}(w) = w - \left(q(w) - \frac{s}{2}\right) = \mathbf{\Delta}(w) + \frac{s}{2}$ and $d_{\text{up}}(w) = \left(q(w) + \frac{s}{2}\right) - w = \frac{s}{2} - \mathbf{\Delta}(w)$ respectively. If $w$ is closest to the upper threshold we have

$$d_{\text{up}} < d_{\text{low}} \implies \frac{s}{2} - \mathbf{\Delta} < \mathbf{\Delta} + \frac{s}{2} \implies \mathbf{\Delta} > 0. \tag{15}$$

While if $w$ is closest to the lower threshold

$$d_{\text{low}} < d_{\text{up}} \implies \mathbf{\Delta} + \frac{s}{2} < \frac{s}{2} - \mathbf{\Delta} \implies \mathbf{\Delta} < 0. \tag{16}$$

We emphasize that this mechanism causes the weight to move towards the quantization thresholds (the edges of quantization "bins") as opposed to the quantization levels (the centers of the quantization "bins"). As shown above, the magnitude of the pull towards the threshold increases as the weight approaches it. The weight will therefore eventually cross the threshold and start oscillating, unless $L(w)$ and $\delta_L$ exactly cancel out, which is unlikely to happen with a finite gradient descent step size.

## 4 Regularization Method

Our theoretical observations in the linear model in Sec. 3, show that the oscillation component is the only part that differentiates QAT from standard full-precision training. We now confirm empirically that the

---

[2]Note that there is no clamping in the quantizer because of the scale factor Eq. (2).

mechanism in Eq. (12) is sufficient to introduce weight oscillations during training of neural networks, and study if this also results in QAT-like behavior with respect to the quantization noise.

From the quantization difference in Eq. (9) and the STE gradient derived in Eq. (12), we have

$$\frac{\partial \mathcal{L}(q(w))}{\partial w} = \frac{\partial \mathcal{L}(w)}{\partial w} - x^2 \mathbf{\Delta}(w),$$ (17)

where the first term is the gradient of the original full-precision loss, and the second term causes the quantization oscillations in QAT.

In order to emulate the effects described in Section 3, we propose a regularization term so that the training objective becomes

$$\mathcal{L}(\mathbf{w}) + \mathcal{R}_\lambda(\mathbf{w}),$$ (18)

where we let the regularization term be similar to the quadratic term in Eq. (9):

$$\mathcal{R}_\lambda(\mathbf{w}) = \frac{\lambda}{2} \sum_\ell \frac{1}{n_\ell} \sum_{i=1}^{n_\ell} \left( q(w_i^\ell)^2 - (w_i^\ell)^2 \right).$$ (19)

Here $\lambda \geq 0$ is a hyperparameter that controls the amount of regularization, $\ell$ ranges over the layers in the model and $i$ over the weights in each layer. In this term, we replaced the factor $x^2$ by a hyperparameter $\lambda$, since the precise expression of $x^2$ is specific to the model studied in Sec. 3. We empirically find that this regularizer is sufficient to induce oscillations. The exploration of the design space of oscillation-inducing regularizers, including layer-dependent and/or adaptive scale factors, is left to future work.

Using the STE, $\frac{\hat{\partial} q}{\partial \mathbf{w}} = 1$, we have the following expression for the gradient:

$$\frac{\hat{\partial}}{\hat{\partial} w_i^\ell} \mathcal{R}_\lambda(\mathbf{w}) = \frac{\lambda}{n_\ell} \left( q(w_i^\ell) - w_i^\ell \right) = -\frac{\lambda}{n_\ell} \mathbf{\Delta}(w_i^\ell).$$ (20)

By the same reasoning as in Sec. 3 this pulls the weight $w_i^\ell$ towards the quantization threshold and causes the gradient to "flip" as $w_i^\ell$ crosses the threshold. We expect this to lead to oscillations based on the same mechanism as in the model from Sec. 3.

Figures 2 and 3 show the results of an experiment where we observe the weight distributions, and measured the oscillations, during training of a neural network (ResNet-18) with varying degrees of regularization, respectively. For comparison, the figures also show the weight distributions and oscillations observed during training with QAT.

Our first observation is that QAT displays more oscillations Fig. 3-(a) than a baseline model without QAT or regularization (corresponding to $\lambda = 0$ in Fig. 3-(b)). As we increase $\lambda$ we observe that the number of oscillations as well as the clustering increases. This confirms that the regularizer from Eq. (19) can indeed induce oscillations similar to QAT during the training of deep neural networks. At $\lambda = 1$ (Fig. 3-(c)) the number of oscillations observed with regularization is similar to the behavior of QAT, lending support to our hypothesis that the mechanism in Eq. (13) is indeed at the root of the oscillations observed when training neural networks with QAT. We use Welch's $t$-test to test if the oscillation counts in Fig. 3, which is the per-weight oscillation count after 50 epochs, are significantly different to the $\lambda = 0$

| Comparison | t-score | p-value |
|---|---|---|
| $\lambda_0$ vs. $\lambda_1$ | 25.13 | < 0.001 |
| $\lambda_0$ vs. $\lambda_{10}$ | 38.86 | < 0.001 |
| $\lambda_0$ vs. QAT | 18.83 | < 0.001 |

Table 1: Welch's t-test comparing the per-weight oscillation count after 50 epochs of training, for the different cases in Fig. 3. We note how the baseline $\lambda = 0$ differs significantly from QAT and regularization induced oscillations with $\lambda = 1, 10$.

baseline, reported in Table 1. Each of the pairwise comparison shows that the distribution of oscillations are significantly different.

## 5   Experiments and Results

In this section, we address the question: *Is inducing weight oscillations during training sufficient to obtain the benefits of QAT?* We answer this question affirmatively using empirical evidence based on the results

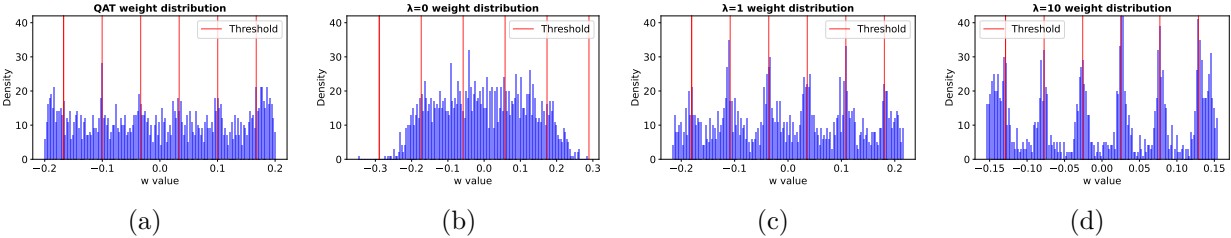

Figure 2: Weight distribution analysis of ResNet-18's first convolutional layer after 50 epochs of training from scratch. a) Weight distribution under QAT with a 3-bit quantizer. b)-d) Our proposed regularization approach with a 3-bit quantizer at varying regularization strengths ($\lambda = 0, 1, 10$, from left to right). When $\lambda = 0$, the training reduces to standard optimization. The QAT distribution (leftmost) exhibits the characteristic threshold clustering behavior. As $\lambda$ increases, we observe progressively stronger clustering of weights around quantization thresholds, illustrating the relationship between regularization strength and weight clustering.

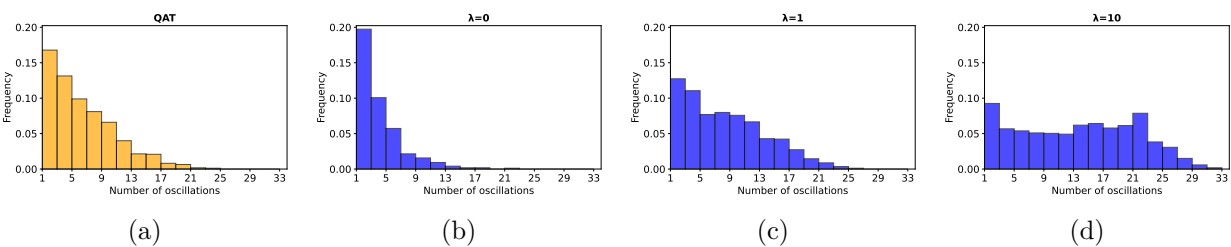

Figure 3: Distribution of weight oscillations. The plots show the distribution of weights with oscillation counts $> 0$ when training with a) QAT and b)-d) the regularizer for different values of $\lambda$. Here $\lambda = 0$ corresponds to a full-precision model where the regularizer has no influence on training. The y-axis represents the percentage of total weights in the first convolutional layer of a ResNet-18 trained from scratch for 50 epochs, while the x-axis shows the oscillation count after 50 epochs. Following the oscillation definition from (Nagel et al., 2022), we count oscillations at each epoch during training. The results demonstrate that QAT produces a significantly higher proportion of oscillating weights compared to $\lambda = 0$. Furthermore, we observe that as we increase $\lambda$ a greater percentage of weights oscillates.

of training ResNet-18 and Tiny Vit on the CIFAR-10 and Tiny-ImageNet datasets. This is both in the training-from-scratch (FS) setting and when fine-tuning (FT) pretrained models.

In the following subsections we first describe the experimental setup, present accuracy results in FS and FT settings for models trained with different quantization levels for the quantizer in $\mathcal{R}_\lambda$ or QAT and finally, and report the cross-bit accuracy of the fine-tuned models. We train models at ternary (three possible values: -1, 0, 1), 3-bit and 4-bit. This is in line with contemporary research, where the emphasis lies on quantization at 4-bit and below, since the challenges of maintaining accuracy are more significant compared to quantization at higher bit widths.

## 5.1 Experimental Setup

We conducted our experiments using the CIFAR-10 (Krizhevsky & Hinton, 2009) and Tiny-ImageNet (Le & Yang, 2015) datasets. We evaluated three architectures; a multi-layer perceptron with five hidden layers and 256 neurons per layer (MLP5), ResNet-18 (He et al., 2016) and Tiny ViT (Wu et al., 2022).

For each architecture we used the Adam optimizer (Kingma, 2014) and tested multiple configurations: a baseline model to establish optimal floating-point (FP) accuracy and PTQ performance, a model with QAT, and a model with regularization using Eq. (19). The two latter configurations are trained using 3-bit and 4-bit quantizers. In all our experiments we use the regularizer $\mathcal{R}_\lambda$ defined in Eq. (19) to induce oscillations (marked as "Oscillations" in the results table).

| Quantization | MLP5 (FS) | ResNet-18 (FS) | Tiny ViT (FT) | ResNet-18 (FT) |
|---|---|---|---|---|
| Baseline FP32 | $51.43 \pm 0.39$ | $83.26 \pm 1.07$ | $96.11 \pm 0.31$ | $88.50 \pm 0.64$ |
| 3-bit PTQ | $20.97 \pm 5.64$ | $77.79 \pm 4.00$ | $11.56 \pm 1.99$ | $10.28 \pm 0.48$ |
| 3-bit QAT | $50.53 \pm 1.43$ | $82.51 \pm 2.14$ | $88.13 \pm 0.60$ | $85.69 \pm 1.83$ |
| 3-bit Oscillations | $48.48 \pm 0.29$ | $81.77 \pm 0.46$ | $88.68 \pm 1.08$ | $84.94 \pm 1.59$ |
| 4-bit PTQ | $46.50 \pm 0.76$ | $82.11 \pm 1.21$ | $21.57 \pm 5.33$ | $35.56 \pm 9.05$ |
| 4-bit QAT | $51.39 \pm 0.60$ | $82.66 \pm 2.57$ | $94.96 \pm 0.33$ | $87.71 \pm 1.14$ |
| 4-bit Oscillations | $50.72 \pm 0.47$ | $83.74 \pm 0.59$ | $94.82 \pm 0.51$ | $87.08 \pm 0.72$ |

Table 2: Performance comparison on CIFAR-10 dataset. Results show classification accuracy for MLP5, ResNet-18, and Tiny ViT across different quantization approaches and bit-widths. Models trained from scratch are marked FS, and fine-tuning experiments are marked FT. FT experiments are based on models pre-trained on ImageNet-1k. In all cases oscillations followed by quantization of the weights matches QAT accuracy. Results are means and standard deviations over 5 random seeds. PTQ results are from the FP32 baseline.

**Training-From-Scratch (FS):** For the MLP5 architecture, we used a learning rate of $10^{-3}$ and regularization parameter $\lambda=1$. The ResNet-18 was trained with a learning rate of $10^{-3}$ and $\lambda=0.75$ (see Appx. A.2 for our hyperparameter selection). We modified the ResNet-18 architecture by replacing the input layer with a smaller $3 \times 3$ kernel and adapting the final layer for 10-class classification of both ResNet-18 and Tiny ViT. Training proceeded for a maximum of 100 epochs with early stopping triggered after 10 epochs without improvement in validation performance. For quantized models, we monitored the quantized validation accuracy at the target bit precision, while for the baseline, we tracked floating-point accuracy.

**Fine-tuning (FT):** We fine-tuned two ImageNet-1k (Deng et al., 2009) pre-trained models on CIFAR-10 and Tiny-ImageNet: a Tiny ViT (learning rate: $10^{-4}$, $\lambda \in \{1, 0.75, 0.5\}$ depending on the bit) and a ResNet-18 (learning rate: $10^{-3}$, $\lambda \in \{1, 0.75, 0.5\}$ depending on the bit). To maintain compatibility with the pre-trained architectures, we up-sampled both CIFAR-10 and Tiny-Imagenet images to $224 \times 224$ pixels. The $\lambda$ parameter selection process for Tiny ViT is detailed in Appendix A.2. Fine-tuning continued for up to 200 epochs on CIFAR-10 and 50 epochs for Tiny-ImageNet, with early stopping after 30 epochs without improvement, using the same accuracy metrics as training from scratch.

**Quantization:** We implemented weight quantization using a per-tensor uniform symmetric quantizer as defined in Eq. (1). PTQ is applied in its most minimal form, by simply quantizing the weights without any calibration. QAT is used as defined in Eq.(5). The quantization range was determined by computing minimum and maximum values per layer. In our implementation of ResNet-18 (11M parameters) all layers except batch normalization were quantized, covering 99.96% of parameters. For Tiny ViT (5.5M parameters) quantization was applied to MLP, Self-Attention, and key-query-value projection layers, encompassing 97.18% of parameters. And lastly for the MLP5 model all layers were quantized. For Tiny-ImageNet models are trained at 3 and 4-bit precision only.

## 5.2 Results

The performance on the two datasets in training-from-scratch and fine-tuning settings is presented in the following sections, along with the observations about cross-bit generalization.

### 5.2.1 Performance on CIFAR-10

**Training from Scratch:** Table 2 (A) shows the results from training an MLP and ResNet-18 from scratch on the CIFAR-10 dataset. Doing only regularization with Eq. (19) demonstrates improvements compared to the PTQ baseline. More importantly, it also matches the performance of QAT at bit widths of 3 and 4.

For both models we see that at 3-bit and 4-bit, using the $\mathcal{R}_\lambda$ regularizer from Eq. (19) exhibits similar performance as QAT but with less variability. With both models, QAT and $\mathcal{R}_\lambda$ regularization are competitive with the full-precision baseline. Notably, both $\mathcal{R}_\lambda$ regularization and QAT significantly outperform PTQ when applied to the full-precision baseline.

| Model | Train bit ↓ / Eval. bit → | FP32 | Ternary | 3-bit | 4-bit | 8-bit |
|-------|---------------------------|------|---------|-------|-------|-------|
| **ResNet-18** | Baseline (PTQ) | 88.50 ± 0.64 | 10.01 ± 0.01 | 10.28 ± 0.48 | 35.56 ± 9.05 | 88.45 ± 0.64 |
| | 3-bit QAT | 16.89 ± 4.97 | 10.01 ± 0.04 | 85.69 ± 1.83 | 17.42 ± 4.96 | 16.56 ± 4.32 |
| | 3-bit Oscillations | **87.86 ± 0.42** | **20.19 ± 10.74** | 84.94 ± 1.59 | **87.56 ± 0.38** | **87.86 ± 0.42** |
| | 4-bit QAT | 87.75 ± 1.13 | 10.13 ± 0.29 | 82.08 ± 6.25 | 87.71 ± 1.14 | 87.76 ± 1.12 |
| | 4-bit Oscillations | 87.85 ± 0.49 | 11.91 ± 0.87 | 85.57 ± 1.10 | 87.08 ± 0.72 | 87.87 ± 0.49 |
| **Tiny ViT** | Baseline (PTQ) | 96.11 ± 0.31 | 9.39 ± 1.11 | 11.56 ± 1.99 | 21.57 ± 5.33 | 96.03 ± 0.34 |
| | 3-bit QAT | 86.94 ± 0.91 | **19.78 ± 6.04** | 88.13 ± 0.60 | 86.69 ± 0.62 | 86.95 ± 0.89 |
| | 3-bit Oscillations | **96.47 ± 0.11** | 9.48 ± 1.64 | 88.68 ± 1.08 | **95.35 ± 0.18** | **96.50 ± 0.11** |
| | 4-bit QAT | 95.14 ± 0.29 | 11.11 ± 1.84 | 59.86 ± 19.95 | 94.96 ± 0.33 | 95.13 ± 0.28 |
| | 4-bit Oscillations | **96.54 ± 0.09** | 11.90 ± 1.29 | 70.23 ± 12.75 | 94.82 ± 0.51 | **96.55 ± 0.09** |

Table 3: Cross-bit evaluation of pre-trained ImageNet-1k models fine-tuned on CIFAR-10. Grey background is the target-bit accuracy. Models are trained using different quantization methods (QAT and ours) and bit-widths (3-bit, and 4-bit), then evaluated across various bit-widths ranging from ternary to FP32. Results are means and standard deviations over 5 random seeds. All significant differences between QAT and Oscillations are shown in bold face.

**Fine-tuning:** Table 2 (B) summarizes the test accuracies for fine-tuning using our $\mathcal{R}_\lambda$ regularization and QAT on ResNet-18 and Tiny ViT architectures with CIFAR-10 and Tiny-ImageNet. The observations are roughly in line with the results observed for training from scratch in the previous section.

For CIFAR-10 as with the case for training from scratch, with both ResNet-18 and Tiny Vit, we see an increase in performance compared to PTQ when using the regularization in Eq. (19). Regularization with $\mathcal{R}_\lambda$ and QAT show comparable performance when quantized at 3 bits and 4 bits, while achieving test accuracy close to the full-precision model at 4-bits.

Performance comparison and related discussions for ternary quantization are presented in Appendix A.3.

**Robustness to Cross-bit Quantization:** In this experiment we evaluated the robustness of oscillations-only and QAT towards quantization at bit widths different from the ones used during training.

When using the $\mathcal{R}_\lambda$ regularization approach, we applied the regularization term with the training bit width during training and applied PTQ after training at a different quantization level. For QAT we trained using the training bit width and afterwards applied PTQ to the latent weights. For each method we also evaluated the corresponding model without PTQ, directly using the latent weights for inference (reported as FP32).

In Table 3 the results from the experiment are reported. A first observation is that the models produced by $\mathcal{R}_\lambda$ regularization consistently achieve nearly full-precision accuracy when quantized at 8-bit or when used without quantization, irrespective of the quantization level used during training. This contrasts with QAT, which produces a viable 8-bit or full-precision model only when trained with at least 4-bit.

Furthermore we see that regularizing using Eq. (19) mostly maintains performance when trained at 3 or 4-bit and quantized at bit level of 3 or 4-bit. QAT also achieves this for Tiny ViT but for ResNet, the accuracy of QAT trained at 3-bit and quantized at other bit widths is barely above random guessing.

Regarding training with ternary quantization, we see that $\mathcal{R}_\lambda$ regularization produces models that achieve near full-precision performance for ResNet when quantized at 3-bit or higher. Ternary training for ViT is somewhat peculiar in that it fails to produce a model that is viable when quantized to ternary, whereas the performance of the resulting models starts to show a high level of variability at 4-bit and finally reaches close to full-precision accuracy at 8-bit. In contrast, for both ResNet and ViT, the performance of QAT degrades completely to random guessing when trained with ternary quantization and evaluated at any other quantization level.

| Quantization | Tiny ViT (FT) | ResNet-18 (FT) |
|---|---|---|
| Baseline FP32 | 67.17 ± 0.67 | 62.93 ± 0.55 |
| 3-bit PTQ | 0.58 ± 0.16 | 0.51 ± 0.03 |
| 3-bit QAT | 44.29 ± 0.49 | 54.08 ± 0.52 |
| 3-bit Oscillations | 44.62 ± 2.47 | 49.34 ± 0.76 |
| 4-bit PTQ | 11.02 ± 2.11 | 20.02 ± 4.80 |
| 4-bit QAT | 60.61 ± 0.16 | 58.31 ± 0.19 |
| 4-bit Oscillations | 60.54 ± 0.37 | 57.26 ± 0.33 |

Table 4: Accuracy on Tiny-ImageNet dataset. Mean and standard deviation is over 3 runs. The models is fine-tuned for 50 epochs on the pretrained ImageNet models. PTQ results is from the FP32 baseline. In both the 3 and 4 bit case, oscillations followed by quantization of the weights matches QAT and is noticeably above the PTQ baseline which has neither oscillations nor QAT.

| Model | Method ↓ / Eval. bit → | FP32 | Ternary | 3-bit | 4-bit | 8-bit |
|---|---|---|---|---|---|---|
| **ResNet-18 (FT)** | Baseline PTQ | 62.93 ± 0.55 | 0.50 ± 0.00 | 0.51 ± 0.03 | 20.02 ± 4.80 | 62.83 ± 0.43 |
| | 3-bit QAT | 50.81 ± 1.85 | 4.51 ± 1.00 | 54.08± 2.39 | 49.76± 1.82 | 50.85± 1.85 |
| | 3-bit Oscillations | **56.67 ± 0.01** | 1.48± 0.10 | 49.34± 0.76 | 55.96± 0.18 | 56.68± 0.03 |
| | 4-bit QAT | 56.57± 1.59 | 0.65± 0.12 | 39.65± 7.33 | 58.31± 0.19 | 56.66± 1.58 |
| | 4-bit Oscillations | **61.58± 0.57** | 0.53± 0.02 | 30.16 ± 4.64 | 57.26± 0.33 | 61.58± 0.52 |
| **Tiny ViT (FT)** | Baseline PTQ | 67.17± 0.67 | 0.49± 0.05 | 0.58± 0.16 | 11.02± 2.11 | 67.06± 0.69 |
| | 3-bit QAT | 39.19± 1.18 | 1.73± 0.05 | 44.29± 0.16 | 36.02± 2.11 | 39.18± 0.69 |
| | 3-bit Oscillations | **56.75± 4.11** | 1.51± 0.86 | 44.62± 2.47 | 56.22± 4.00 | 56.78± 4.07 |
| | 4-bit QAT | 59.75± 0.73 | 0.49± 0.02 | 34.42± 1.93 | 60.61± 0.16 | 59.73± 0.80 |
| | 4-bit Oscillations | **65.58± 0.29** | 0.54± 0.09 | 22.26± 4.76 | 60.54± 0.37 | 65.60± 0.31 |

Table 5: Cross-bit evaluation of pre-trained ImageNet-1k models fine-tuned on Tiny-ImageNet. Grey background is the target-bit accuracy. Models are trained using different quantization methods (QAT and ours) and bit-widths (3-bit, and 4-bit), then evaluated across various bit-widths ranging from ternary to FP32. Results are means and standard deviations over 5 random seeds. All significant differences between QAT and Oscillations are shown in bold face.

### 5.2.2 Performance on Tiny-ImageNet

**Fine-tuning:** Table 4 summarizes the test accuracies for the Tiny-ImageNet dataset. Here we observe the same tendency as with CIFAR-10; oscillations provides a significant increase in accuracy compared to the PTQ baseline. While for the Tiny ViT model $\mathcal{R}_\lambda$ regularization is sufficient to recover the quantized accuracy of QAT in both the 3 and 4-bit case, for ResNet18 there is a degradation in accuracy at 3-bit.

**Robustness to Cross-bit Quantization:** In Table 5 we see the cross-bit results from the Tiny-ImageNet experiments. As with CIFAR-10 we note that the models produced by $\mathcal{R}_\lambda$ regularization achieves a better cross-bit performance at bits higher than the target bit. Though we do note a change in the cross-bit behavior; the cross-bit results for 3 and 4-bit is generally lower and not as close tot he FP baseline as in the CIFAR-10 case, yet still there is a significant difference between QAT and $\mathcal{R}_\lambda$ regularization.

## 6 Discussion

We have shown that training with weight oscillations induced via $\mathcal{R}_\lambda$ regularization is sufficient in most cases to maintain performance after quantization for ResNet and Tiny ViT. The primary effect of our regularizer is to push weights toward quantization thresholds, which results in weights that are robust to quantization. This "threshold-pushing" dynamic naturally leads to two observable phenomena: weight clustering and oscillations. This begs the question whether weight oscillations are also a necessary part of the QAT training process. Indeed, some previous work already points towards this. There are examples claiming that both dampening and/or freezing of oscillations too early in the training process is detrimental to performance after quantization (Nagel et al., 2022; Han et al., 2021). And in other case presented in Liu et al. (2023), freezing

only the low frequency oscillating weights improves performance. This suggests that weight oscillations are both a necessary and sufficient part of QAT, at least in the early phases of the training process. This further supports our hypothesis that oscillations in QAT have a positive effect on quantization robustness overall.

Yet we do note deviations from QAT when regularizing with Eq. (19): QAT outperforms $\mathcal{R}_\lambda$ regularization at ternary quantization (Appendix A.3), whereas $\mathcal{R}_\lambda$ regularization outperforms QAT in cross-bit accuracy for the ternary and 3-bit case. In A.5, we see how it seems that the cross-bit performance for QAT is upper-bounded by the target-bit performance, which might explain the subpar QAT performance at cross-bit compared to $\mathcal{R}_\lambda$ regularization which seems bounded by the full-precision accuracy.

**Limitations and Future Work**  Our theoretical analysis was performed using the toy model in Section 3, and the regularization term is motivated using this analysis. We expect other effects that are not entirely captured by this analysis to be part of QAT. This is explored further in Appendix A.1, where we show how the second term in Eq. (9) is not zero in the gradient when there are multiple layers.

The second term in Eq. (9) is closely related to the oscillations-dampening methods such as the one presented in Equation 6 in (Nagel et al., 2022), which works by adding a linear pull towards the quantization level. In Appendix A.1 we show how for the linear case with two weights, the second term is no longer zero in the gradient and as such QAT then consists of two components; one that creates oscillations and one that dampens them. From this we can see that oscillations alone do not capture the full dynamics of QAT as analyzed in linear models.

**Broader Impact:**  The work presented here investigates the oscillations that arise when preparing deep neural networks for quantization using QAT. By characterizing these oscillations, we gain deeper insight into the dynamics of QAT, which can be leveraged to enhance its effectiveness and, consequently, the efficiency of quantized models. Improved efficiency translates into reductions in memory, computation, and energy demands, facilitating the deployment of high-performing quantized networks on low-power edge devices. Some of these applications can have broader positive impact (for example, medical electronics) and also negative impact (for example, surveillance systems or weapons).

## 7  Conclusions

Based on the analysis of a linear model we hypothesized that weight oscillations during training in deep neural networks make the model robust to quantization akin to QAT. In Sections 3 and 4 we explain how training with QAT and STE leads to oscillations and propose a regularizer that encourages this oscillating behavior. We confirm that as we increase the strength of the regularization, and empirically observe the appearance of clustering together with oscillations.

Finally, we experimentally confirm that the regularizer indeed leads to consistent robustness towards quantization for 3-bit and 4-bit quantization levels. Our oscillations by regularization approach achieves comparable performance to QAT above ternary quantization when quantizing to the target-bit seen during the optimization. Furthermore, we also observe that it shows increased robustness compared to QAT in cross-bit quantization with quantization levels higher than the target-bit used in the quantizer during training. All this is evidence in favor of our hypothesis.

Our insights on weight oscillations and their role in quantization robustness open new horizons for model quantization approaches which usually build on the idea of aligning weights at quantization levels – the opposite of what seems to be the core dynamic in QAT. The regularization approach especially creates interesting possibilities for cross-bit robustness, potentially making the regularization method more appealing than QAT when the goal is to deploy or release a single set of model weights that can work across different bit widths or maybe even quantizers. While the regularizer used in our experiments should be viewed as an initial step, we expect that quantization robustness could be further improved by developing oscillation-inducing methods that are adaptive to different learning rates, layer statistics or phases of the training process.

**Acknowledgments:** Authors thank Tong Chen, Jákup Svøðstein, Sophia Wilson, Sebastian Hammer Eliassen, Shivam Adarsh and members of SAINTS Lab for useful discussions throughout.

JW, BP and RS are partly funded by European Union's Horizon Europe Research and Innovation Action programme under grant agreements No. 101070284, No. 101070408 and No. 101189771. RS also acknowledges funding received under Independent Research Fund Denmark (DFF) under grant 4307-00143B.

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

## A  Appendix

### A.1  Two-layer with Single Weights

Consider a linear model $f(x) = w_2 w_1 x$, with weights $w_1, w_2$, input $x$, and target $y \in \mathbb{R}$. The quantized version of this model is defined as $f_q(x) = q(w_2)q(w_1)x$, where $q(\cdot)$ is the quantizer from Eq. Equation (1). The quadratic loss for the model is given by

$$\mathcal{L}(f(x)) = \frac{1}{2}\big(w_2 w_1 x - y\big)^2$$

The difference compared to full-precision optimization is then given as

$$\delta_{\mathcal{L}} = \mathcal{L}(f_q(x)) - \mathcal{L}(f(x)) \tag{21}$$

$$= \frac{1}{2}\Big[\big(q(w_2)q(w_1)x - y\big)^2 - \big(w_2 w_1 x - y\big)^2\Big] \tag{22}$$

$$= \frac{1}{2}\Big[\big(q(w_2)q(w_1)x\big)^2 - \big(w_2 w_1 x\big)^2 - 2y\big(q(w_2)q(w_1)x - w_2 w_1 x\big)\Big] \tag{23}$$

$$= \frac{1}{2}x^2\Big[q(w_2)^2 q(w_1)^2 - w_2^2 w_1^2\Big] + yx\Big[w_2 w_1 - q(w_2)q(w_1)\Big] \tag{24}$$

The loss difference decomposes into:

$$\underbrace{\frac{1}{2}x^2\Big(q(w_2)^2 q(w_1)^2 - w_2^2 w_1^2\Big)}_{\text{Quadratic term (Oscillator)}} \quad + \quad \underbrace{yx\Big(w_2 w_1 - q(w_2)q(w_1)\Big)}_{\text{Linear term (Oscillation Dampener)}} \tag{25}$$

Taking the derivative of $\mathcal{L}$ w.r.t $w_1$:

$$\frac{\partial \delta_{\mathcal{L}}}{\partial w_1} = \frac{\partial}{\partial w_1}\Big(\mathcal{L}(f_q(x)) - \mathcal{L}(f(x))\Big) \tag{26}$$

$$= \frac{\partial}{\partial w_1}\left[\frac{1}{2}x^2\Big(q(w_2)^2 q(w_1)^2 - w_2^2 w_1^2\Big) + yx\Big(w_2 w_1 - q(w_2)q(w_1)\Big)\right] \tag{27}$$

$$= x^2\left[q(w_2)^2 q(w_1)\frac{\partial q(w_1)}{\partial w_1} - w_2^2 w_1\right] + yx\left[w_2 - q(w_2)\frac{\partial q(w_1)}{\partial w_1}\right] \tag{28}$$

Using the STE approximation from Eq. 5, we get:

$$\frac{\partial \hat{\delta}_{\mathcal{L}}}{\partial w_1} = x^2\left[q(w_2)^2 q(w_1) - w_2^2 w_1\right] + yx\left[w_2 - q(w_2)\right] \tag{29}$$

We note that the linear term is no longer zero in the gradient and thus for a model consisting of two single weight layers we see that there are additional effects from QAT other than oscillations. Furthermore, we would like to clarify that even though the two layered model under consideration does not have explicit non-linear activation functions, it cannot be reduced to a single layer model. This is because of the non-linearity of the rounding operation.

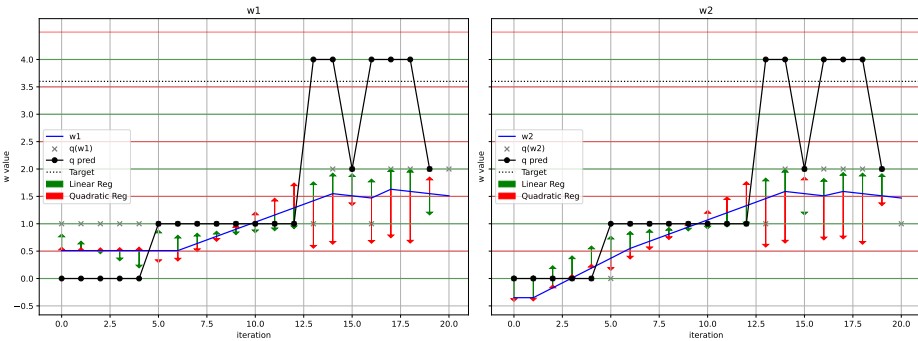

Figure 4: We repeat the toy model experiments, but this time with two weights, taking into account that the linear term is no longer zero in the gradient. We notice at epoch 15 and 18 where the prediction of the quantized model is greater than y, the effect of the terms flip for $w_2$.

## A.2 Hyperparameters

### A.2.1 ResNet-18

In Fig. 5 and Fig. 6 we see the results of the $\lambda$ hyperparameter search over different learning rates for a ResNet-18 model. There is a clear trend of seeing the best performance at a learning rate of $10^{-3}$. We note that interestingly there is a comparable performance for a wide range of $\lambda$s, indicating that it is the presence of oscillations which is important for quantization robustness, and not the exact frequency of oscillations.

### A.2.2 Tiny ViT

In Fig. 7 we note how also the Tiny Vit seems to allow for a wide range of $\lambda$s even though we this time note that $\lambda = 1$ performs significantly better than the others.

## A.3 Ternary Quantization

Performance comparison for the ternary quantization for different models on CIFAR-10 is reported in Table 6. While both QAT and oscillations improve the PTQ baseline significantly, oscillations degrade compared to QAT, especially for the Tiny ViT. This is in line with previous literature (Liu et al., 2023), where transformers

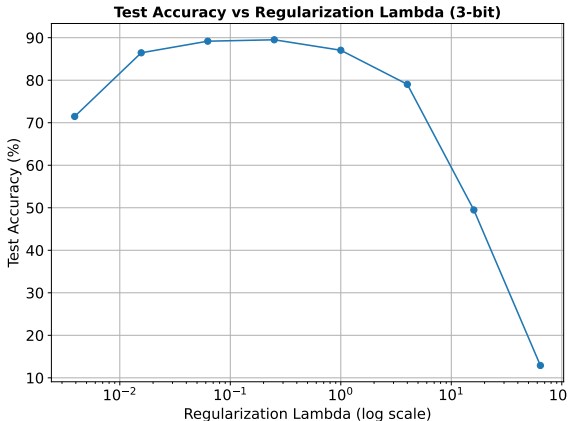

Figure 5: Mean over 3 runs of the best test accuracy for different lambdas. Fine-tuning a pretrained ResNet-18 on CIFAR-10 for 50 epochs. Quantizer is set to 3-bit and $10^{-3}$ learning rate and 100% of the training data is used.

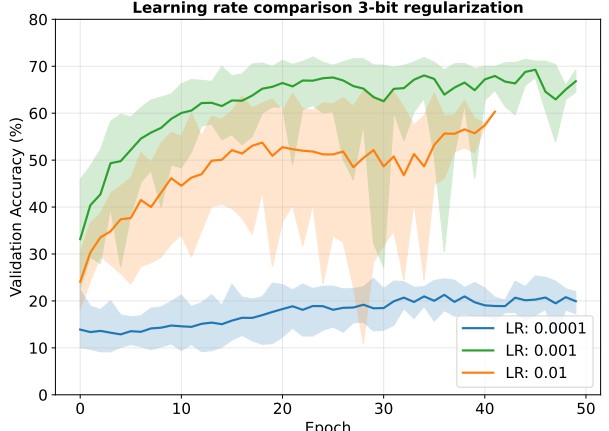

| $\lambda$ | 3-bit (%) | Ternary (%) |
|---|---|---|
| 0.25 | $68.77 \pm 0.19$ | $47.85 \pm 5.51$ |
| 0.50 | $69.47 \pm 1.11$ | $46.77 \pm 4.83$ |
| 0.75 | $70.08 \pm 0.40$ | $46.86 \pm 3.01$ |
| 1.00 | $66.20 \pm 4.05$ | $47.33 \pm 2.06$ |
| 1.25 | $69.31 \pm 0.32$ | $43.14 \pm 6.62$ |
| 1.50 | $68.96 \pm 0.30$ | $46.73 \pm 3.91$ |
| 1.75 | $69.92 \pm 0.11$ | $47.02 \pm 4.19$ |

Figure 6: Mean over 3 runs of the best validation accuracy for different lambdas. Training a ResNet-18 from scratch. Both ternary and 3-bit is at $10^{-3}$ learning rate and 50% of the data used for training. The plot shows three learning rates, where we evaluate with the $\lambda$s for each learning rate (shown in the table on right). The colored background covers the range between the maximum and minimum value of the quantized validation accuracy with the given $\lambda$s.

have been identified as especially sensitive to oscillations. If oscillations are fundamental to QAT, this still leaves the question of why QAT achieves superior ternary performance. Appendix A.1 offers an indication of the underlying reason. Looking at the gradient of the two-layer toy model, in Eq. 25 we note that the second term is no longer zero in the gradient. This term bears close resemblance to existing formulations of oscillation dampeners, such as in Eq. 6 of Nagel et al. (2022). These works by pushing weights towards their nearest quantization level, thereby dampening or nullifying the effect of any oscillator (which pushes weights towards their nearest threshold). We therefore speculate that the main component of QAT is still oscillations, but that QAT also has an inherent dampening mechanism. Given that in a uniform quantizer, the quantization error increases exponentially as we decrease the bits in the quantizer, ternary quantization's large error magnitude makes the absence of dampening particularly detrimental, resulting in sub-optimal quantized performance compared to QAT.

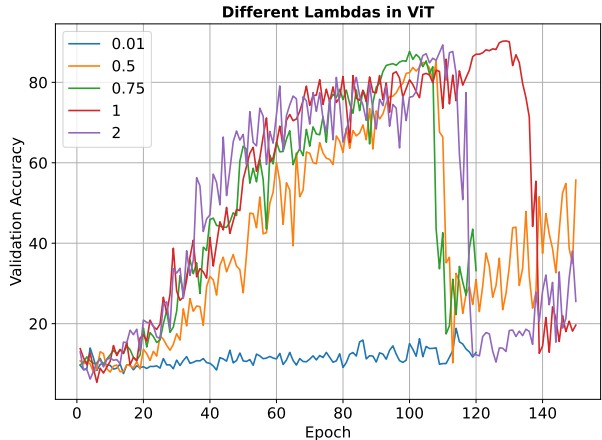

| $\lambda$ | 3-bit (%) | Ternary (%) |
|------|------------|-------------|
| 0.01 | 18.85 | - |
| 0.5 | 85.21 | 15.10 |
| 0.75 | 87.68 | - |
| 1.0 | 90.29 | 13.04 |
| 2.0 | 89.31 | 14.16 |
| 2.5 | - | 13.70 |
| 5.0 | - | 14.20 |

Figure 7: Validation accuracy at different $\lambda$ values and the corresponding best validation accuracies for 3-bit and 2-bit configurations for a single run. Learning rate is set to $10^{-4}$ for fine-tuning. For the 2-bit we test higher $\lambda$ but still see no improvement in accuracy. We note how all the $\lambda$s lies close to each other, except for the low of $10^{-2}$.

| Quantization | MLP5 (FS) | ResNet-18 (FS) | ResNet-18 (FT) | Tiny ViT (FT) |
|---|---|---|---|---|
| Baseline FP32 | $51.43 \pm 0.39$ | $83.26 \pm 1.07$ | $83.26 \pm 1.07$ | $96.11 \pm 0.31$ |
| Ternary PTQ | $10.00 \pm 0.02$ | $10.00 \pm 0.01$ | $10.00 \pm 0.01$ | $9.39 \pm 1.11$ |
| Ternary QAT | $49.20 \pm 1.34$ | $79.62 \pm 6.42$ | $77.02 \pm 7.57$ | $73.53 \pm 0.77$ |
| Ternary Oscillations | $36.49 \pm 0.51$ | $61.50 \pm 1.82$ | $44.59 \pm 3.30$ | $13.51 \pm 1.32$ |

Table 6: Performance comparison with ternary quantization on CIFAR-10 dataset. Mean and standard deviation is over 3 runs. The models is fine-tuned for 50 epochs on the pretrained ImageNet models. PTQ results is from the FP32 baseline. For both oscillations only and QAT we see a significant improvement over the PTQ baseline. Yet oscillations degrade significantly compared to QAT, especially for the Tiny Vit. FS: Train from scratch. FT: Fine-tuned.

## A.4 Epochs and cross-bit robustness

There is an interesting interaction between number of epochs trained and robustness both of our method and QAT. We note how QAT converges first for the target-bit and then over time also converges for the 4 and 8-bit. Additionally we see that QAT seems upper-bounded by the target-bit performance, while this is not the case for oscillations only as shown in Fig. 8.

## A.5 Convergence behavior during oscillations-only optimization

Fig. 9 shows the convergence behavior of the full precision weights and the quantized weights at target-bit. We note how the Tiny ViT displays a peculiar convergence behavior, where the accuracy breaks, only to go up again. In the Tiny Vit model we regularize the self-attention layers. It is already noted in Liu et al. (2023) that ViTs are especially vulnerable to oscillations in the query and key of self-attention layers, which might be related to the convergence behavior observed when regularizing with Eq. 19.

## A.6 Weight Distribution and Oscillation Frequency

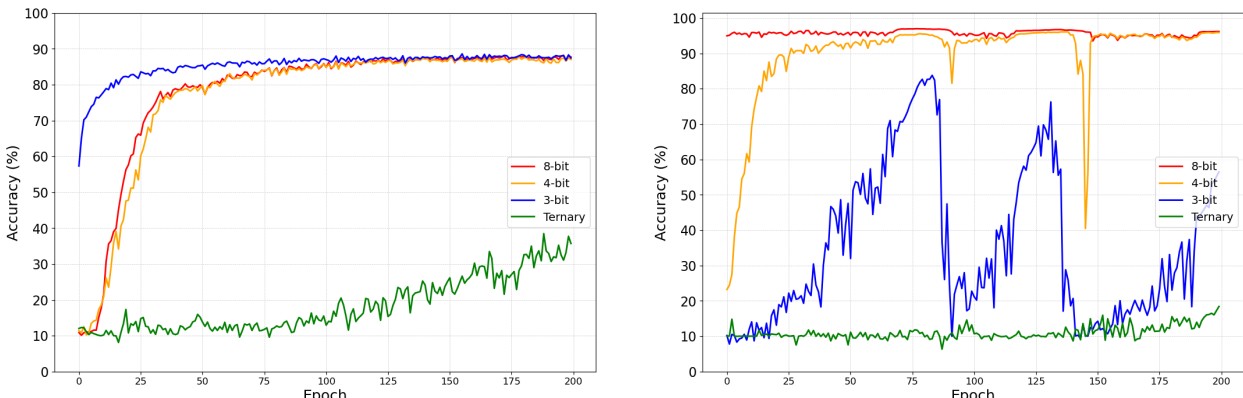

Figure 8: Left is the validation accuracy during training of a ViT with QAT at different bits, right is for our regularization. Both QAT and regularization is trained with a 3-bit quantizer. We note how the order of convergences for cross-bit changes between QAT and our model and that QAT cross-bit robustness especially depends on number of epochs trained.

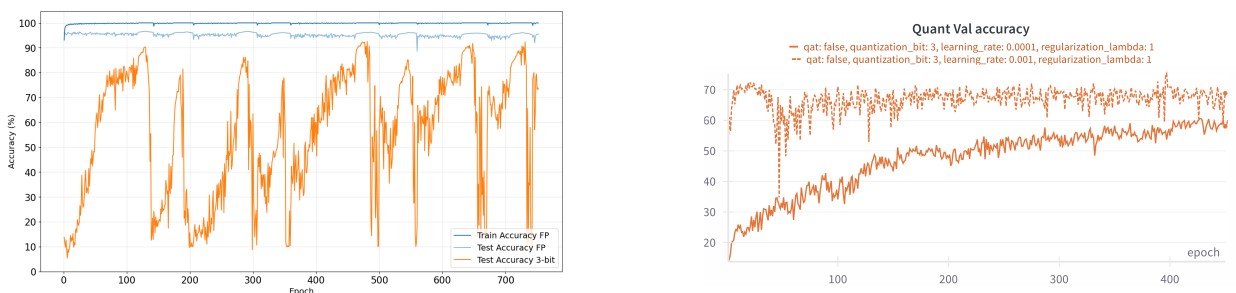

Figure 9: (right) Convergence behavior of ResNet-18. (left) Convergence behavior of a Tiny ViT with regularization with a 3-bit quantizer. Note the peculiar behavior of the orange line, which is the validation accuracy on the target-bit performance. The performances cycles between ≈ 90% and 10%, while the full precision accuracy (the model evaluated without quantized weights) stays some-what stable.

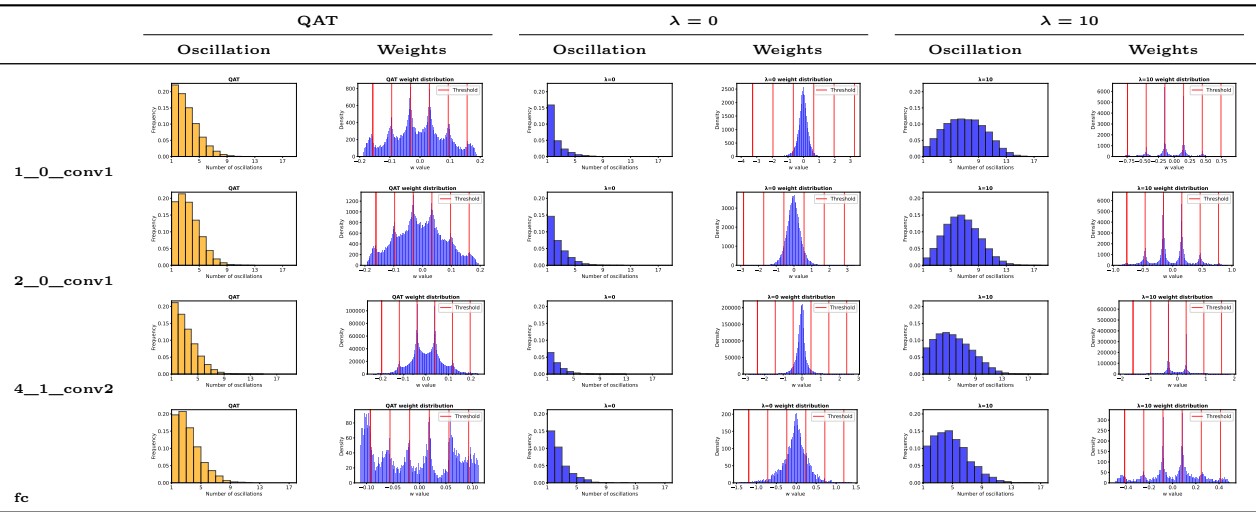

Figure 10: Oscillations count and weight distribution side by side across layers and regularization settings. Each pair (left: oscillation, right: weights) corresponds to the same $\lambda$ configuration. Note that weight distribution plots do not share x and y axis.

