# OpenReview forum: "Oscillations Make Neural Networks Robust to Quantization"
_TMLR — Accepted by TMLR_

### Review · Reviewer_pwtX · 2025-09-05

**Summary Of Contributions:**

This work shows that the oscillation effects in QAT training, contrary to conventional wisdom, can be beneficial

**Additional Comments:**

This review is short because I do not see major problems with this paper, even the figures are pretty nicely done

**Audience:**

Yes

**Audience Explanation:**

This is a standard  machine learning / deep learning problem, so yes

**Claims And Evidence:**

Yes

**Claims Explanation:**

The experiments are rather extensive

The derivations are informal and to-the-point and are easy to understand

**Requested Changes:**

Nothing that I spot

---

> ### Author Response · Authors · 2025-10-24
> **Response to Reviewer pwtX**
>
> We are very pleased and encouraged by this positive feedback from the reviewer. We appreciate the reviewer's excitement about our work. Thank you.

---

### Review · Reviewer_xy6A · 2025-09-08

**Summary Of Contributions:**

This paper studies quantization aware training (QAT), and in particular investigates the empirical evidence that weights tend to "oscillate" around a quantization threshold during training. The purpose of this paper is to give a sense to these oscillations, and to stand with the perspective that they are a natural (or even beneficial) aspect of QAT. To do so, the Authors analyze a univariate linear regression model to explain this behaviour analytically. Then, from the derivation they semi-heuristically define a regularization term that could be added to the loss to recreate the bias of QAT. Finally, the authors investigate empirically the effects of such regularizer, finding positive results on a discrete variety of tasks.

**Audience:**

Yes

**Audience Explanation:**

This work look into QAT, which is a primary sub-area of research in deep learning. This work brings solid evidence to clarify the empirical phenomenon of oscillations during such training methods, and I therefore presume many individuals in the community would appreciate this reading.

**Broader Impact Concerns:**

No concerns

**Claims And Evidence:**

Yes

**Claims Explanation:**

The paper is well written, self contained, and the technical results are correct. The numerical evidence seems solid.

**Requested Changes:**

**Minor**

- STE as it is introduced in the first section might be a bit obscure to people not in the field (it was to me). Maybe it is worth it to just skip mentioning it (it does not seem so necessary at this point) or to add few words to describe what the algorithm look like, adding a cross reference to its formal definition later in the paper (same comment goes for the mention at point 1 of the contributions).

- Right after Eq. (3), $\Delta$ is natural to be interpreted as a vector, if one thinks about a model with more than one parameter being quantized. Then the notation $| \Delta |$ after this equation is a bit "sloppy". Maybe the Authors can replace it with $| \Delta_i |$ to mean each entry of $\Delta$, and remark that $q$ is meant to be taken component-wise?

- As somebody that was not aware of STE, I struggled a bit at first in understanding the algorithm. In particular, the fact that at every iteration $w$ is not a quantized parameter, as it is stored as "continuous" all along, and the quantization is used only in the forward passes during training and to release the final model (is this correct?). Can the Authors perhaps add few more line of explanation about this?

- Maybe the authors can remark at the level of the contributions that the toy model is univariate. This is a quite simplifying setting, and maybe it is fair to stress it before.

-  The proposed regularizer is in form different from what would give rise to the gradient difference in your toy model (Eq. (18) does not come directly from Eq. (8) if I understood directly). If this is indeed the case, can the Authors remark that this definition is heuristic?

---

**Major**

In the course of the paper, it seems there is remarked some sort of causal connection between "oscillations" and successful quantization. For example, see the sentence in the first paragraph of Section 6 "weight oscillations are both a ncessary and sufficient part of QAT", or (more directly) the title of the paper.

Possibly, my concern stems from the fact that an "oscillation" is not exactly defined in the paper. For this reason, the empirical connection between successful quantization, clustering around the thresholds, and oscillations (as measured for the plots in Figure 3) remains qualitative, and it is not obvious to state that "oscillations are good for quantization", as there are other latent variables here.

I am talking from the perspective of somebody that does not have much experience in this sub-field of research, so it might be possible that this nomenclature of things is simply well accepted by the community. However, I still believe the Authors could be more careful in drawing this connection, or at least to elaborate on it in the Section 6.

---

> ### Author Response · Authors · 2025-10-24
> **Response to Reviewer xy6A**
>
> We thank the reviewer for their careful reading of our paper and for the constructive and detailed feedback. We appreciate the assessment of our work and the suggestions for improvement.
>
> **Minor**
>
> * **Clarification of STE:** We will clarify the background section on STE. We now include a subsection in Sec. 2 which details the STE algorithm:
>
>     > STE is a heuristic gradient approximation technique commonly used to enable backpropagation through functions $f(\cdot)$ whose gradients are zero almost everywhere. In the forward pass, the function  $f(\mathbf{x})$ is applied as usual. During the backward pass, however, its gradient is approximated by replacing the derivative of $f$ with the constant 1. Formally, STE defines the approximate gradient as $\frac{{\partial} f}{{\partial} \textbf{x}} \approx \frac{\hat{\partial} f}{\hat{\partial} \textbf{x}} = 1.$
>
>     In the background seection on STE we will start the section with:
>
>     > In Quantization-Aware Training (QAT), the network maintains full-precision weights for learning but quantizes them during the forward pass before computation. This allows training to proceed with gradients applied to full-precision weights, while the network effectively operates on quantized values."
>
> * We agree that the notation can be more precise and will correct it to emphasize that it is taken componentwise. We will update after equation (3) with:
>     > "... absolute error of each component is bounded between $0 \leq |\Delta_i| \leq \frac{s}{2}$"
>
>     And before Equation (1):
>     > We consider an elementwise uniform scalar quantizer with a ...
>
> * We will update Contribution 1 to reflect that it is a univariate model. This is updated in the abstract and in other relevant places.
>
> * In Contribution 2, we mention that the regularizer is inspired by the toy-model analysis; we can make this clearer if necessary. We also discuss in Section 4 why we chose the specific formulation of the regularizer.
>
> **Major**
>
> * **Formalization of Oscillations:** We will add a formal, quantitative definition of an “oscillation” in Section 2, based on the measurement criterion we used (from Nagel et al., 2022). We have introduced a new subsection in Sec. 2 that defines oscillations now.
>
>     > We use the definition from Nagel (2022) which defines a weight oscillation during QAT to occur in iteration \(t\) if it satisfies both the following conditions:
>     >
>     > 1. The quantized value of the weight needs to change between iterations i.e. $q(w_t) \neq q(w_{t-1})$.
>     > 2. The direction of the change in the quantized domain needs to be the opposite than that of its previous change i.e. $\mathrm{sign}(\Gamma_t) \neq \mathrm{sign}(\Gamma_\tau)$, where $\tau$ is the iteration of the last change, and  $\Gamma_t = q(w_t)  - q(w_{t-1})$ is the direction of the change.
>
> * **Connection between quantization, clustering, and oscillations:** We agree with the reviewer that these causal claims may be too strong, so we will revise the discussion to emphasize that oscillations are a symptom of the (beneficial) underlying dynamics, not necessarily the cause. Specifically, we will state:
>     > The primary effect of our regularizer is to push weights toward quantization thresholds, which results in weights that are robust to quantization. This “threshold-pushing” dynamic naturally leads to two observable phenomena: weight clustering and oscillations.

---

> > ### Comment · Reviewer_xy6A · 2025-11-03
> >
> > I thank the Authors for their rebuttal and for their action in implementing the suggested details. In particular, I appreciate the *Weights Oscillations* paragraph in Section 2 and I am glad for the minor edit in the Abstract about univariate linear regression.

---

### Review · Reviewer_qnLH · 2025-10-14

**Summary Of Contributions:**

This paper argues that the performance improvements seen with Quantization Aware Training (QAT) result from oscillations that QAT induces in the training dynamics. This is argued through analyzing the difference in gradients, in a 1D linear model, that one observes with versus without QAT. This is further supported by experiments where the authors introduce a regularization term during regular training (i.e. without QAT) that causes similar oscillations to occur and that leads to similar performance results as QAT. The authors also note that the oscillations appear, in certain cases, to lead to more robust generalization than QAT across a range of different quantizations that the models were not trained on.

## Strengths
I find the main strengths of this paper to be the results, and presentation. The result that oscillations seem to reproduce some key features of quantization is quite interesting, and the approach of using regularization to show this in experiments seems novel and informative. Similarly, the result that oscillations alone lead to better cross-bit generalization in many cases is interesting and seems like it could have useful applications. In terms of presentation, the structuring of the paper is largely easy to follow, and most of the figures are easy to understand.

## Weaknesses
I find four main weaknesses with the paper. I think that they should be able to be resolved with edits and a little bit more work.
1. My main concern with the work is that certain aspects of the results seem to be explored in a cursory fashion. In particular, it would be nice to see more than just the input layer data in Fig 1, 3. See Requested changes “elaboration on certain results”, below, for more details.
2. It would be nice to have some measures of statistical significance for a subset of the results. See “statistical significance” section below for more details.
3. Portions of the background material could be presented a little more clearly. See “need for elaborated background” section below for more details.
4. There are some issues of clarity with the writing, including grammar, spelling, and weird wording issues, at a number of points. This is important to resolve, but I think this should be fairly easily addressed. See “clarity” section below.

Additionally: there are some more minor issues with tables and figures that I mention in the “extra table and figure issues” section.

**Additional Comments:**

**Important:** Note that, while I have a strong deep learning background, I am not very familiar with quantization. For this reason I cannot comment on how this work fits into the broader literature, but only on the soundness of the results and the clarity of the exposition. Please take this into account when considering my review.

**Audience:**

Yes

**Audience Explanation:**

Quantization is an exciting frontier for improving computational benefits of machine learning models, which has potential for high impact particularly in light of growing model size. This paper provides some exciting results about quantization aware training with the straight through estimator. For this reason I think the paper is of interest to a subset of the TMLR community.

**Broader Impact Concerns:**

It could be useful to touch on both positive and negative impacts of this work that stem from potential improvements to quantization of neural networks. To give some examples: in terms of benefits, quantization improvements seem likely to improve computational efficiency, potentially reducing energy footprints. This could be useful given the huge energy consumption of large deep learning models. In terms of downsides, improvements in computational efficiency could result in larger models being implemented on smaller devices. This could be applied, for example, to drones used for military and surveillance purposes, resulting in human rights infringements and violence.

**Claims And Evidence:**

No

**Claims Explanation:**

I believe that most of the paper's claims are well supported. I have only answered "no" here because of two reasons that are mentioned in the requested changes, below. Specifically, I would (1) like to see data in Fig 2 and 3 on more than just the first layer of the ResNet, and (2) like to see statistical tests on Fig 3 showing that the difference in oscillations is significant.

**Requested Changes:**

I have organized the requested changes into sections based on the main weaknesses. I have added “**C**” at the end of changes that I view as critical, for my review, and “**S**” after ones that I view more as suggestions.

# Elaboration on Certain Results
- In Fig.2, 3 the plotted data is from the first convolutional layer of a ResNet. Why the first layer? Do you only find these results to hold in the first layer? If so, perhaps it is because of the single layer analysis in the 1D model. Perhaps comment on this, and, critically, please also show data from other layers too. **C**
- Second last paragraph of Discussion: “where we show how the second term is not zero”. It’s not clear here what the second term you are referring to is. Could be useful to elaborate. **C**
- Appendix: “Additionally because of the non-linearity of the rounding operation, even with the absence of a non-linear activation function, we can no longer reduce the model to a single weight.” What do you mean by this? **C**
- Appendix, Figure 7: why does validation accuracy drop at around 120 epochs? Does oscillation make learning unstable? Should discuss this **C**

# Statistical Significance
- Fig 2, Fig 3: would be nice to include some test of statistical significance to compare whether the displayed distributions are significantly different from each other. Otherwise cannot say that, e.g., QAT leads to significantly here oscillations than other conditions **C**
- Table 2: where are the SDs in this table? Could be useful to include statistically significant differences. At the very least, SDs should be shown **C**

# Need for Elaborated Background
- (intro, paragraph 3): Straight-Through Estimator (STE) mentioned for the first time. Could be worth explaining in brief. **S**
- (section 2): could you explain whether the floating point weight value is maintained during training with QAT, or whether, after each weight update, the current weights are set to the closest quanta? **C**
- (section 2 paragraph 2): what is a max-range scale factor? If this is defined in Eq.2, maybe mention eq.2 earlier? **S**
- (section 2 paragraph 3): what is the “usual clamping operation in the quantizer”? **S**
- (3rd paragraph of “Related work” section in section 2): what does “at individual bits” mean here? **C**
- (footnote on page 4): could you elaborate on what clamping means here? **C**

# Clarity
- weird wording (intro, paragraph 1): “grown increasingly powerful at many tasks” => grown increasingly successful at solving difficult modelling tasks” **S**
- grammar/wording (intro, paragraph 2): the first sentence in this paragraph seems grammatically suspect. Could maybe solve with a colon perhaps **C**
- word choice (intro, paragraph 2): “degradations” => “reductions” **C**
- writing flow: (intro, paragraph 3): “quantization” mentioned 3 times in first sentence; suggest replacing with another word for better flow of text **S**
- redundancy: (intro, paragraph 3): “Unfortunately, QAT brings its own mysteries” seems redundant. For example, could be removed and replaced by starting the next sentence with “however”. **C**
- redundancy: (intro, paragraph 3): “In particular, QAT often exhibits a seemingly strange phenomenon: oscillations in the weights.” => “In particular, QAT often exhibits oscillations in the weights during optimization, while regular training does not.” **C**
- vague wording (intro, paragraph 4): “What if we had this view on oscillations in QAT backwards?” => “what if QAT oscillations are actually beneficial for training”. If this change was made, one would likely want to re-write the paragraph a bit **S**
- writing issues (section 2 paragraph with “Quantization-Aware Training” heading): by “fundamentally” do you mean “typically”? Also, in the same paragraph, you don’t need to say “variants of QAT” twice in the first sentence. **C**
- redundancy (3rd paragraph of “Related work” section in section 2): this paragraph seems to repeat stuff mentioned earlier in the intro. Could be worth checking to see if redundancy can be removed **S**
- redundancy (3rd paragraph of “Related work” section in section 2): “In short, …” => remove. **S**
- redundancy (equation 6): this can be removed **C**
- (third last paragraph of section 5.2.1): it is written “In Table 6 shows the results from the experiments”. Note the grammar error. **C**
- Second last paragraph of Discussion: “is not zero in the gradient, when there are multiple layers” => remove comma. **C**
- Second last paragraph of Conclusions: “all this being evidence of our hypothesis” => “all this is evidence in favour of our hypothesis”. **S**
- Acknowledgements section: you will probably want to update this section. **S**
- Appendix: “we see that there is additional effects from QAT” => “we see that there are additional effects from QAT” **C**
- spelling issue (figure 6 caption): “for each learning ratue evaluate” **C**
- spelling (figure 7 caption): “but still see no improvemenet in accuracy” **C**

# Extra Table and Figure Issues
- Third last paragraph of section 5.2.1: it is written “In Table 6 shows the results from the experiments”. Shouldn’t this be “table 2? Regardless, table 2 does not seem to be referenced in the body of the paper. **C**
- Last paragraph of section 5.2.1: should ref table 6 here I think. **C**
- Appendix, Figure 4: is "target" just "y"? and "q pred" the predicted value? If so, maybe label them "y" and "y predicted" or something like this. **S**
- Appendix: “In Fig. 5 and Fig. 6 we see the results of the λ hyperparameter search over different learning rates for a ResNet-18 model. There is a clear trend of seeing the best performance at a learning rate of 10−3.” Here, it would be nice for the order of figures to match the order of mention in the text. I.g. should switch the order of the figures or should mention regularization tuning before learning rate. **S**
- Should put tables in appendix that are not referenced in the appendix either all at top or at bottom, to avoid mixing them up with the appendix text, in my opinion. **S**

---

> ### Author Response · Authors · 2025-10-24
> **Response to Reviewer qnLH**
>
> We thank the reviewer for their meticulous reading of our work. Both the critical changes (**C**) and other suggestions (**S**) have helped improve our manuscript. We also refer to the general response in relation to the changes made in the revision. We collect the reviewer comments into common points and address them below:
>
> * **Oscillations from other than first layer:** The oscillations are observed across layers; we initially showed only the first layer for brevity. We have included a new figure in the Appendix which shows the oscillations across different layers, which expand on the Figures 2 and 3.
> *  **Statistical significance of oscillation frequencies:** We use Welch’s $t$-test to test if the oscillation frequencies in Fig. 3 is significantly different to the $\lambda=0$ baseline. These are now reported in a new Table 1.  We observe that the baseline $\lambda=0$ differs significantly in oscillation frequency both from QAT and regularization induced oscillations with $\lambda=1,10$.
> *  >> Second last paragraph of Discussion: “where we show how the second term is not zero”. It’s not clear here what the second term you are referring to is. Could be useful to elaborate. C
>
>
>     We have now clarified this by adding added: "...in Eq. 9 is not zero in the gradient when there are multiple layers"
> *  >>Appendix: “Additionally because of the non-linearity of the rounding operation, even with the absence of a non-linear activation function, we can no longer reduce the model to a single weight.” What do you mean by this? C
>
>     We have clarified this by: "Furthermore, we like to clarify that even though the two layered model under consideration does not have explicit non-linear activation functions, it cannot be reduced to a single layer model. This is because of the non-linearity of the rounding operation."
>
> * >> Appendix, Figure 7: why does validation accuracy drop at around 120 epochs? Does oscillation make learning unstable?
>
> 	We already address this in the appendix section "Convergence behaviour during oscillations-only optimization".
>
> * >> Table 2: where are the SDs in this table? Could be useful to include statistically significant differences. At the very least, SDs should be shown.
>
>     The standard deviation values were reported in the Appendix tables. Now we have added standard deviations to all accuracy tables in the main text, and removed the SD tables in the appendix sections.
>
> * **Need for elaborated background:** We have clarified the points raised by the reviewer in the revised text. For example, we now provided a dedicated subsection that describes STE in Sec. 2, redefined the quantizer to not use min-max values (by using $\max(|\mathbf{w}|)$), and clarified the text around _clamping_.
> * **Clarity, Extra Table and Figure Issues:** All the main points have been addressed and marked in red.

---

> > ### Comment · Reviewer_qnLH · 2025-10-26
> > **response to author response #1**
> >
> > Thank you very much to the authors for the many updates. In particular, I’m very excited to see the multi-layer data and the T-tests. I have a few extra comments that I’ve arranged in bullet points below:
> > - Just to confirm, the t-test is on the total number of oscillations observed in a given trained network on a given training run, right? If this is the case, it might be useful to spell this out in a bit more detail. In the current description the authors use oscillation “frequency” to describe what is being measured but, if I am not mistaken, it is the mean number of oscillations not just per unit time but also per weight in the network that is being investigated. While one could call this a frequency, it is a little confusing since frequency is usually cycles/unit time for a waveform rather than (potentially sparse) cycles/(unit time + network size).
> > - On page 6 you mention: “Our first observation is that QAT displays more oscillations
> > - Fig. 2-a)”. I think that you mean figure 3-a here, right? If so, please do check other figure references because incorrect references were a problem in my first reading of the paper.
> > - Related to the above: if you first mention oscillation frequency/counts and then mention clustering second, please put the oscillation frequency/counts plot first.
> > - Fig 3 (similar to the first bullet point): is the x-axis number of oscillations per training run? If so, it could be worth labelling it “number of iterations per training run” just to be clear. It also might be worth labeling the y-axis something different than frequency, to avoid people confusing this with frequency of oscillation.
> > - Table 5: could be worth mentioning “same format as figure 3”, or something like this, just to be clear.
> > - Page 13: “Furthermore, we like to clarify…” => Furthermore, we would like to clarify
> > - Figure 10: “Note that weight distribution plots does not share x and y axis.” => Note that weight distribution plots do not share x and y axis.
> > - Lastly, I just want to follow-up about the broader impact of the paper. I mentioned this near the end of my review but, unless I am mistaken, I don’t believe the authors engaged with these comments.
> >
> > Note: I just want to highlight how, in just checking updated sections of the paper, I found a couple grammar/spelling problems and potential figure reference issues. I have not combed through the entire paper again, in favour of focusing on the updated sections, but I encourage the authors to do so, to make sure that there aren’t small errors like this that might find their way into a published version of the paper.
> >
> > Thanks again for the many great updates!

---

> > > ### Author Response · Authors · 2025-10-27
> > > **Response to rebuttal**
> > >
> > > We appreciate the continued engagement and careful feedback by the reviewer. We have not updated the pdf, as the other reviewers are engaging with the previous update. We address the comments below:
> > >
> > > > Just to confirm, the t-test is on the total number of oscillations observed in a given trained network on a given training run, right? If this is the case, it might be useful to spell this out in a bit more detail ...
> > >
> > > We agree that the wording for Table 1 was unclear; we will clarify to state that it is the oscillation count and not frequency.
> > >
> > > As already stated in Fig. 3: "The y-axis represents the percentage of total weights in the first convolutional layer of a ResNet-18 trained from scratch for 50 epochs, while the x-axis shows the oscillation count"
> > > We will change the wording from frequency to count at the end of Section 4 and under the table for figure 1 (t-tests).
> > > For Table 1:
> > > "Welch’s t-test comparing the per-weight oscillation count after 50 epochs of training, for the different cases in Fig. 3. We note how the baseline λ = 0 differs significantly from QAT and regularization induced oscillations with λ=1,10."
> > >
> > > And the text at the end of section 4 to:
> > > "We use Welch’s t-test to test if the oscillation counts reported in Fig. 3, which is the per-weight oscillation count after 50 epochs, are significantly different from the λ = 0 baseline, reported in Table 1. Each of the pairwise comparisons shows that the distribution of oscillations is significantly different to λ = 0."
> > >
> > > > On page 6 you mention: “Our first observation is that QAT displays more oscillations Fig. 2-a)”. I think that you mean figure 3-a here, right? If so, please do check other figure references because incorrect references were a problem in my first reading of the paper.
> > >
> > > We will correct this to point to Fig. 3-a.
> > >
> > > > Related to the above: if you first mention oscillation frequency/counts and then mention clustering second, please put the oscillation frequency/counts plot first.
> > >
> > > We will adjust this according to the suggestion by first discussing the clustering points, and then oscillation counts.
> > >
> > > > Fig 3 (similar to the first bullet point): is the x-axis number of oscillations per training run? If so, it could be worth labelling it “number of iterations per training run” just to be clear. It also might be worth labeling the y-axis something different than frequency, to avoid people confusing this with frequency of oscillation.
> > >
> > > As noted in answer 1, we already describe the axes: "The y-axis represents the percentage of total weights in the first convolutional layer of a ResNet-18 trained from scratch for 50 epochs, while the x-axis shows the oscillation count" To make it more clear we will change the last line to:
> > > "... x-axis shows the oscillation count after 50 epochs". Additionally we will mention this in the description of the new figure 10 as well.
> > >
> > >
> > >
> > > > Table 5: could be worth mentioning “same format as figure 3”, or something like this, just to be clear.
> > >
> > > We will fix the caption to point this out.
> > >
> > > > Page 13: “Furthermore, we like to clarify…” => Furthermore, we would like to clarify
> > > > Figure 10: “Note that weight distribution plots does not share x and y axis.” => Note that weight distribution plots do not share x and y axis.
> > > > Note: I just want to highlight how, in just checking updated sections of the paper, I found a couple grammar/spelling problems and potential figure reference issues. I have not combed through the entire paper again, in favour of focusing on the updated sections, but I encourage the authors to do so, to make sure that there aren’t small errors like this that might find their way into a published version of the paper.
> > >
> > > We will adjust these sentences, and check the language carefully throughout the paper.
> > >
> > > > Lastly, I just want to follow-up about the broader impact of the paper. I mentioned this near the end of my review but, unless I am mistaken, I don’t believe the authors engaged with these comments.
> > >
> > > We apologise for the oversight. We simply missed updating this in our previous response. We will add the following based on the reviewer's suggestion.
> > >
> > > #### Broader Impact:
> > > The work presented here investigates the oscillations that arise when preparing deep neural networks for quantization using QAT. By characterizing these oscillations, we gain deeper insight into the dynamics of QAT, which can be leveraged to enhance its effectiveness and, consequently, the efficiency of quantized models. Improved efficiency translates into reductions in memory, computation, and energy demands, facilitating the deployment of high-performing quantized networks on low-power edge devices. Some of these applications can have broader positive impact (for example, medical electronics) and also negative impact (for example, surveillance systems or weapons).

---

> > > > ### Comment · Reviewer_qnLH · 2025-11-03
> > > > **final reviewer comment**
> > > >
> > > > Thank you to the authors for addressing all my comments! If they implement all the changes that they have outlined then I'm quite satisfied by their responses and associated updates.

---

### Author Response · Authors · 2025-10-24
**Overall response to all reviews**

We thank all three reviewers for their useful feedback and overall positive assessments. We address the comments from each of the reviewers in individual responses. We present a summary of the high-level changes in this general comment.

* All _significant_ changes in response to reviewer comments are highlighted in red in the manuscript.
* Suggestions for language refinement have been incorporated; these are marked in red as well. We do not provide point-by-point response to these changes.
* We have restructured the Appendix for better clarity by moving some content from the main text into the Appendix, and vice versa.
* New analyses as requested by some of the reviewers are included in the Appendix.

We have updated the PDF with these changes. The suggestions by all reviewers have improved our manuscript considerably. We appreciate their careful reading our work.

---

### Decision · Action_Editor_vyPC · 2025-11-24

**Recommendation:** Accept with minor revision

**Additional Comments:**

Dear Authors,

For the minor revision instructions, please, just update the paper to the camera ready format and update the text color. Given that you included all changes and finalized the paper during the discussion phase, I don't see any other needed changes (I have marked a minor revision only because of that).

**Audience:**

Yes

**Audience Explanation:**

Quantization is one of the key topics in ML especially in the era of large models and LLMs (both for training and inference footprint, and on-device inference). The paper investigates the weight oscillations observed during Quantization Aware Training (QAT) and shows that this effect is an essential part of QAT. Better understanding of mechanisms happening during QAT are of high interest for the community and the paper's findings could open new research directions in the quantization domain.

**Claims And Evidence:**

Yes

**Claims Explanation:**

The paper investigates effect of weights oscillation in Quantization Aware Training (QAT), and shows that training benefits from this oscillation (in contrary community before was considering this as a poor effect): authors start from simple univariate linear regression model to explain this behaviour analytically, and then authors introduce regularization term that induces similar oscillations into training dynamics and in the end leads to better training for variety of tasks. Experiments are done on CIFAR-10, Tiny ImageNet with MLP, ResNet, and Tiny-ViT.

All reviewers agreed that the paper is well written, simple to follow, technical results are correct, experiments and analysis are very extensive. The main feedback for the paper was about improving further readability by adding minor clarifications, disambiguating places even for people outside the domain, fixing typos and grammar, and being more precise in formalism. The only extra results reviewers asked for are including all layers analysis and statistical significance.

Authors reworked all comments from reviewers and in the end there were no further requests from reviewers for the latest revision. All reviewers recommend acceptance of the paper.

---

> ### Author Response · Authors · 2025-11-26
> **Decision and Camera ready version**
>
> Dear AE and Reviewers
>
> We are extremely pleased with the decision. We thank your efforts again in helping us improve our submission.
>
> All the final changes have been incorporated and the camera ready version is now posted.
>
> Kind regards,
>
> All the authors.